# Coral growth, survivorship and return-on-effort within nurseries at high-value sites on the Great Barrier Reef

**Lorna Howlett[1,2], Emma F. Camp** ◉ [1]*, **John Edmondson[2], Nicola Henderson[1], David J. Suggett[1]**

**1** Climate Change Cluster, Faculty of Science, University of Technology Sydney, Ultimo, NSW, Australia,
**2** Wavelength Reef Cruises, Port Douglas, QLD, Australia

* Emma.Camp@uts.edu.au

**Data Availability Statement:** All relevant data are within the manuscript and its Supporting Information files.

## Abstract

Coral reefs are deteriorating worldwide prompting reef managers and stakeholders to increasingly explore new management tools. Following back-to-back bleaching in 2016/2017, multi-taxa coral nurseries were established in 2018 for the first time on the Great Barrier Reef (GBR) to aid reef maintenance and restoration at a "high-value" location–Opal Reef–frequented by the tourism industry. Various coral species (n = 11) were propagated within shallow water (ca. 4-7m) platforms installed across two sites characterised by differing environmental exposure–one adjacent to a deep-water channel (Blue Lagoon) and one that was relatively sheltered (RayBan). Growth rates of coral fragments placed onto nurseries were highly variable across taxa but generally higher at Blue Lagoon (2.1–10.8 cm$^2$ month$^{-1}$ over 12 months) compared to RayBan (0.6–6.6 cm$^2$ month$^{-1}$ over 9 months). Growth at Blue Lagoon was largely independent of season, except for *Acropora tenuis* and *Acropora hyacinthus*, where growth rates were 15–20% higher for December 2018-July 2019 ("warm season") compared to August-December 2018 ("cool season"). Survivorship across all 2,536 nursery fragments was ca. 80–100%, with some species exhibiting higher survivorship at Blue Lagoon (*Acropora loripes*, *Porites cylindrica*) and others at RayBan (*A. hyacinthus*, *Montipora hispida*). Parallel measurements of growth and survivorship were used to determine relative return-on-effort (RRE) scores as an integrated metric of "success" accounting for life history trade-offs, complementing the mutually exclusive assessment of growth or survivorship. RRE scores within sites (across species) were largely driven by growth, whereas RRE scores between sites were largely driven by survivorship. The initial nursery phase of coral propagation therefore appears useful to supplement coral material naturally available for stewardship of frequently visited Great Barrier Reef tourism (high-value) sites, but further assessment is needed to evaluate how well the growth rates and survival for nursery grown corals translate once material is outplanted.

**Funding:** Essential funding for the work was provided from the AMP Foundation (Tomorrow Maker Award to DJS), "Future-proofing the Great Barrier Reef through climate change-resilient super corals", and the Australian & Queensland Governments ("Solving the bottleneck of reef rehabilitation through boosting coral abundance: Miniaturising and mechanising coral out-planting" to DJS, EFC, JE). Additional contribution of EFC to manuscript writing and final preparation was through the University of Technology Sydney Chancellor's Postdoctoral Research Fellowship and ARC Discovery Early Career Research Award (DE190100142). Wavelength Reef Cruises provided support in the form of salaries for authors JE and LH but did not have any additional role in the study design, data collection and analysis, decision to publish, or preparation of the manuscript. The specific roles of these authors are articulated in the 'author contributions' section.

**Competing interests:** JE is co-owner and LH is an employee of Wavelength Reef Cruises. This does not alter our adherence to PLOS ONE policies on sharing data and materials. There are no patents, products in development or marketed products associated with this research to declare.

## Introduction

Deterioration in global coral reef health has prompted intensive efforts to explore and implement interventions that can enhance existing management efforts [1–3]. Passive interventions include the restoration of environmental conditions, such as improvement of water quality, to allow for the recovery of reef systems whereas active restoration efforts range from the construction of artificial reefs to coral transplantation [4–7]. Intervention approaches aimed at enhancing coral abundance have focused on general ecosystem recovery in response to physical disturbance, attempting to repair reef structural damage caused by ship groundings in Florida [4], and blast fishing or extreme weather/climate in the Indo-Pacific [8], but also to restore populations of a particular species or genus, such as *Acropora* spp. decimated by disease throughout the Caribbean and western Atlantic [9, 10]. Until recently, such interventions had not been applied to the Great Barrier Reef (GBR). However, climate change has increased the intensity and frequency of marine heat waves to the extent where >30% of all coral was lost on the GBR in 2016/17 alone [11]. Many GBR "high-value reef sites", in particular those generating high economic revenue via the tourism industry [12], face reduced coral abundance and rates of natural recovery [13].

Many, if not all, current coral transplantation approaches rely on first propagating coral populations to build biomass faster than natural recruitment allows, through either sexual reproduction or asexual fragmentation [1, 14]. Coral nurseries have become increasingly adopted across reef sites worldwide to continually propagate ("farm") corals [15], with a means to isolate environmental growth conditions [16] and/or to track specific coral genotypes [17]. Nurseries have also been established to house larval propagules or even "fragments of opportunity" (fragmented coral available on site) prior to outplanting [18]. Numerous examples of nurseries have been reported from the Caribbean [10, 17, 19], Red Sea [20, 21], and Indo-Pacific [16, 22, 23], employing many different engineering solutions. Nursery structures have included either free floating mid-water platforms [20, 21, 23, 24], or frames directly fixed to reef-neighbouring substrates [10, 16, 19, 22]. Fragments from these nursery designs must then be manually removed from the nursery for outplanting. Consequently, rope-based mid-water nurseries have also been developed [23, 25], which have the potential to be directly attached to the reef and so reduce the need for outplanting of individual fragments. Similarly, frames have been designed to be fastened directly onto the reef and sown for propagation to bypass a need for fragment outplanting [4].

Given the broad variety of approaches used to propagate corals using nurseries, it is perhaps unsurprising that the immediate yields–a product of growth and survivorship [26]–within nursery settings are highly variable as a result of different environmental conditions [24], and alternate coral species [20, 24] or genotypes [10, 17]. Even so, species propagated within nurseries appear to exhibit growth rates and survivorship similar to, or exceeding, those of source colonies within natural reef habitats [10, 15, 20, 22, 24]. However, despite these growing reports of nursery-based propagation outcomes, alternative engineering approaches and propagation contexts have rarely been quantitatively compared across the various efforts to date (e.g. *Acropora cervicornis* in the Caribbean, [10]). As such, it is not fully resolved how different factors govern yields when using different approaches, sites and species, thereby constraining capacity to initially optimise new propagation practices, such as those recently implemented on the GBR [15, 26].

Whilst naturally occurring fragments of opportunity are desirable for coral outplanting at high-value tourism sites on the GBR [27], sites with reduced live coral cover rarely have a continuous supply required for routine site maintenance (J.E., Pers. Obs.), in particular where using high-throughput practices can regularly result in hundreds of corals outplanted per day

[27]. We therefore installed the first multi-taxa coral nursery on the northern GBR in 2018 (Opal Reef; [26]) to evaluate "success" of growing coral on coral nurseries. Here, we first present growth and survivorship from the first 9- and 12-month periods for various coral species maintained within platform nurseries installed at two sites on Opal Reef. We then apply parallel growth and survivorship measurements to determine "success" within the nurseries across taxa and sites according to a novel scoring method to describe relative return-on-effort (RRE, [26])–the RRE is a score obtained from corresponding measures of growth and survivorship for any given taxa/environment. As such the RRE complements mutually exclusive assessments of growth or survivorship as "measures of success", to consider how growth and survivorship interact for any given species, e.g. differences in species with "r versus k" life history strategies that carry inherent trade-offs in growth rates and stress tolerance [26]. In doing so, we show for the first time how nurseries can yield variable growth rates of 0.6–10.8 cm$^2$month$^{-1}$ and high survivorship of $> 80\%$ at two frequently visited tourism (high-value) sites on the GBR, and that local differences in site location, and presumably quality of environmental conditions for optimum growth, inevitably influence these yields.

## Materials and methods

All fragmentation, assembly and deployment of nurseries was conducted under permit G18/40023.1 issued by the Great Barrier Reef Marine Park Authority.

Two nursery sites were located on Opal Reef (16°13'S 145°53'E), ca. 50 km from Port Douglas, Queensland, within the Great Barrier Reef Marine Park (Fig 1). Opal Reef is a 24.7 km$^2$ table-top reef with a sheltered, sandy lagoon adjacent to the coast, becoming progressively deeper towards the outer points of the reef flat. Opal Reef has high economic value as it is easily accessible and consequently experiences high intensity visitation by tourism operators [12]. A total of 17 commercial moorings are situated around the entire Opal Reef, which is currently split into two zones for commercial and public use by the Great Barrier Reef Marine Park Authority (GBRMPA): the Conservation Park Zone, allowing limited hook-and-line fishing, boating, snorkelling and diving, and the Marine National Park Zone where fishing activities are prohibited. Whilst manta tow surveys around Opal Reef suggest that the 2016/17 coral bleaching event reduced hard coral cover from 21% (2015) to 8% (2019) [28], coral mortality from this event was highly patchy within and between sites (J.E., Pers. Obs.). For example, hard coral cover varies from 17.1% to 39.8% (based on replicate 30m continuous line intercept transects conducted in October 2018; S1 Table). Opal Reef is therefore a prime location for localised restoration via nursery-based "coral farming". Two sites were chosen at Opal Reef for installation of nurseries given their high accessibility from routine tourism operations for maintenance and monitoring.

The first nursery site, RayBan (RB) (16°13'27"S 145°53'22"E), was installed in February 2018 and located in a Marine National Park Zone (Fig 1). RB is a shallow, central, protected area of Opal Reef where the reef-scape is scattered with coral outcrops and sandy lagoons, with the nursery area at a depth of ca. 6m and within 5m of nearby coral outcrops. The second nursery site, Blue Lagoon (BL) (16°12'18"S 145°53'54"E), was installed in August 2018 and located within a Conservation Park Zone (Fig 1) on a sandy bottom at a depth of ca. 8m within 10m of a coral outcrop. BL is subject to tidal currents via close proximity to a deep-water channel leading into the Coral Sea. In contrast, RB is rarely subjected to tidal currents due to its sheltered location.

Environmental data has not been collected at these sites, and temperature loggers installed with the initial nursery frames unfortunately failed. We therefore extracted remotely sensed sea surface temperature (SST) from the GIOVANNI online system for satellite-derived data

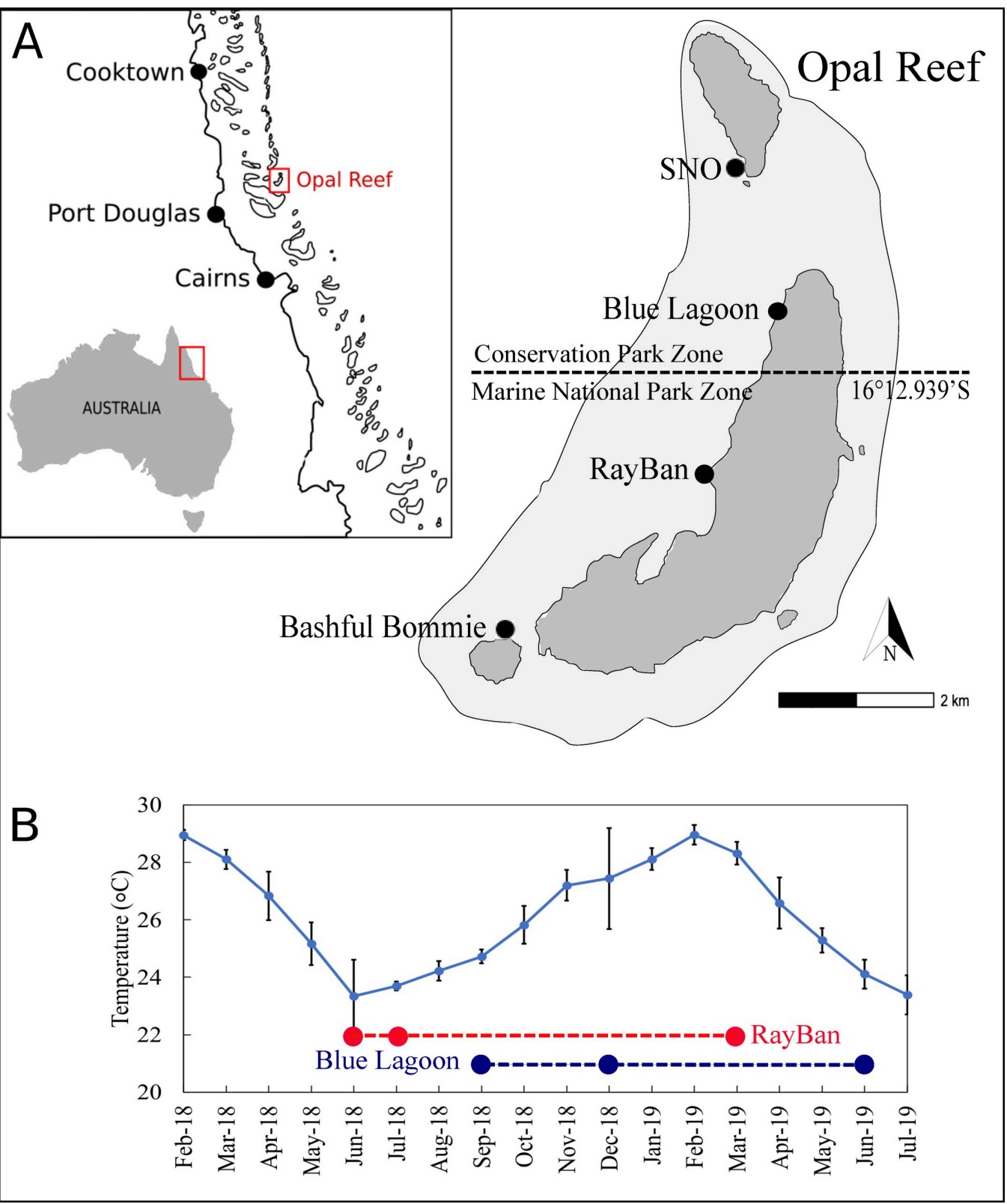

**Fig 1.** **(A)** Location of sites on Opal Reef, Great Barrier Reef (Australia), including nursery sites RayBan (RB; 16˚13'27"S 145˚53'22"E) and Blue Lagoon (BL; 16˚12'18"S 145˚53'54"E) in relation to their location within the region and country. Delineation of the management zones for Opal Reef (see main text) is indicated by the dashed line. **(B)** The remotely sensed mean (± standard error, n = 3–4 per month) Sea Surface Temperatures (SST) for Opal Reef. Imposed onto the SST are points showing the time when measurements were taken for both RayBan and Blue Lagoon nurseries.

maintained by NASA (http://disc.sci.gsfc.nasa.gov/giovanni), using monthly area-averaging bounded to 145.86W, -16.25N, 145.91E, -16.17N (and hence capturing the entire study area around Opal Reef of ca. 30km$^2$) between February 2018 and July 2019. The data used was collected by the Moderate Resolution Imaging Spectroradiometer (MODIS-aqua) for night SST (8 days, 4km).

Each nursery frame consisted of 2 x 9kg Besser blocks placed on the sea floor and attached via spliced rope to an aluminium diamond-mesh frame 2.0 x 1.2m, supported by a 20L float (S1 Fig). Frames were suspended approximately 1-2m above the substrate to reduce potential sedimentation so that corals were held at 4-5m and 6-7m depth for RB and BL nurseries, respectively. All aluminium frames were conditioned *in situ* for a period of at least 2 weeks prior to seeding with coral.

All coral fragments were sourced from the reefs neighbouring the nursery area (<100m) and at the same depth (±1m) as the frames. Source colonies for any given species were sampled randomly throughout the local population (spaced at least 5m from one another). Fragments placed on the RB and BL nurseries were sourced from a total of 60 and 26 colonies, respectively (Table 1), and are referred to as "source" (or parent) colonies, where a maximum of 10% of each colony was removed using a hammer and chisel. Source material was further fragmented on-deck using either a hammer and chisel or pliers, resulting in fragments for the nursery spanning a broad range of sizes (0.33 to 91.2 cm$^2$, maximum length x width). The fragments were attached via cable ties to the conditioned aluminium frames on-deck in a purpose built tray, in which there was a continuous flow of filtered seawater to reduce the air exposure time for each fragment. All coral fragments were kept under shade for the entirety of this process. A total of 1,440 and 1,096 fragments were ultimately attached across 7 and 4 nursery frames for RB and BL, respectively, and spanning 11 coral species: *Acropora humilis*, *Acropora hyacinthus*, *Acropora intermedia*, *Acropora loripes*, *Acropora millepora*, *Acropora tenuis*, *Montipora hispida*, *Montipora spumosa*, *Pocillopora* cf. *verrucosa*, *Porites cylindrica*, and *Turbinaria reniformis* (Table 1). The various coral species were chosen given their relatively high abundance within the total coral cover on the neighbouring reef and survival through the recent 2016/17 heat waves [26]. However, the ultimate number of fragments for any one taxon seeded onto the frames was also determined by ease of fragmentation; consequently, species such as *A. tenuis*, which were highly abundant and typically yielded >50 fragments per colony, resulted in

**Table 1. Summary of coral species used to seed nursery frames at Opal Reef sites Blue Lagoon (BL; August 2018) and RayBan (RB; May 2018).**

| Species | Morphology | Source colonies (n) | | Nursery frags. (n) | |
|---|---|---|---|---|---|
| | | BL | RB | BL | RB |
| *Acropora hyacinthus* | Plating | 5 | 5 | 216 | 103 |
| *Acropora humilis* | Corymbose | 4 | 5 | 159 | 116 |
| *Acropora intermedia* | Branching Open | 1 | | 19 | 19 |
| *Acropora loripes* | Corymbose | 1 | 7 | 38 | 210 |
| *Acropora millepora* | Corymbose | 2 | 10 | 85 | 183 |
| *Acropora tenuis* | Corymbose | 6 | 9 | 372 | 493 |
| *Montipora hispida* | Encrusting Long Upright | 1 | 2 | 53 | 26 |
| *Montipora spumosa* | Encrusting | | 4 | | 57 |
| *Pocillopora* cf. *verrucosa* | Branching Closed | 4 | 10 | 81 | 122 |
| *Porites cylindrica* | Encrusting Long Upright | 2 | 2 | 73 | 31 |
| *Turbinaria reniformis* | Laminar | | 6 | | 80 |
| **TOTAL** | | | | **1096** | **1440** |

Morphology was classified as per Precoda et al. [30].

more fragments on the nursery. Once attached, each and every fragment was assigned an identifier code based on their position on the assigned frame. A random subset of 406 (RB) and 240 (BL) fragments from across all species were assessed for tracking growth on the frames over time during opportunistic visits from the tourism operations vessel. All fragments were monitored for survivorship, and any dislodgement of loose fragments from nursery frames recorded as "loss" (<2–3% across taxa) and not considered in counts of survivorship. Initial size of the monitored fragments was determined using scaled photographs through image analysis (ImageJ2; [29]) to retrieve the maximum length and width. Parallel measurements using callipers (ca. 25–50 measurements per taxa) were used to validate the image-retrieved length and width.

Aluminium frames were placed *in situ* upon complete seeding with fragments, and the nursery assembled underwater using SCUBA. All seeding for this study was conducted in May and August 2018 –and hence at the beginning and end of the "cool season" (see Fig 1) for RB and BL, respectively. No manual cleaning of the aluminium frames was required since platforms remained largely algal free via grazing by herbivorous fish from adjacent coral outcrops (S1 Fig), an observation generally consistent with recent reports from nurseries elsewhere [31]. All platforms were monitored on a regular basis (every 30–60 days) to assess survivorship, with any dead fragments immediately removed. Furthermore, any fragments with visible signs of tissue loss were removed (as per permitting conditions to pre-empt potential disease outbreaks) and accounted for as "loss" under survivorship; as such, any size measurements inherently contained 100% tissue cover. To assess growth, maximum length and width of the tracked fragments, fragments were re-measured using callipers via SCUBA in December 2018 and July 2019 at BL. Few fragments were removed from the BL nursery for outplanting during this period enabling growth and survivorship to be compared for the winter-run up to summer ("cool season", August—December), versus summer-run down to winter ("warm season", January—July), as well as for the entire year (Fig 1). In contrast, fragments for RB were only remeasured in August 2018, three months after installation, and February 2019 (Fig 1), with many fragments removed for outplanting during this period. As such, we compare growth rates and survivorship for RB versus BL here from different deployment periods (9 versus 12 months, respectively; Fig 1), and return to this point later.

Growth rates were determined as change in size (areal extension, length x width) over time ($\Delta G$; cm$^2$ month$^{-1}$) and survivorship as the proportion of all initial fragments remaining over time (%). In order to further evaluate success, we subsequently determined the relative growth rates as % growth month$^{-1}$ (= $[\Delta G/G_I] \cdot 100$), where $G_I$ is initial size, in addition to assessing the life-history strategy trade-off of growth versus survivorship via RRE [26]. Normalising $\Delta G$ to $G_I$ provides a means to compare growth with previous studies where different units for growth have been used [26]. Importantly, we did not observe any size dependency of % growth month$^{-1}$ on $G_I$ for our dataset here (S2 Fig). RRE was determined for each fragment based on corresponding values of % growth versus % survivorship. All percentage data was first tested for normality (Shapiro–Wilk) and subsequently arcsine (% Survivorship/100) or Ln transformed (% Growth) to stabilise variance. RRE was then assigned as a score (between 0 and 20) from a scoring matrix bounded by Ln (% Growth) values -2 to +8 versus arcsine (% Survivorship/100) values 0–1.57, with higher RRE scores indicating higher $\Delta G$ and survivorship [26].

A series of analysis of variance (ANOVA) with post hoc Tukey tests were undertaken to compare absolute growth, percent growth (that was Ln transformed) and RRE at BL (August 2018-July 2019). Two-way ANOVAs with post hoc Tukey tests were used to compare the same parameters between sites, BL and RB, and also temperature ("warm" versus "cool" season) just at BL. Tests for normality (qq-plots) and equal variance (Levene's test) were passed. Statistics were run in RStudio version 1.1.423 [32].

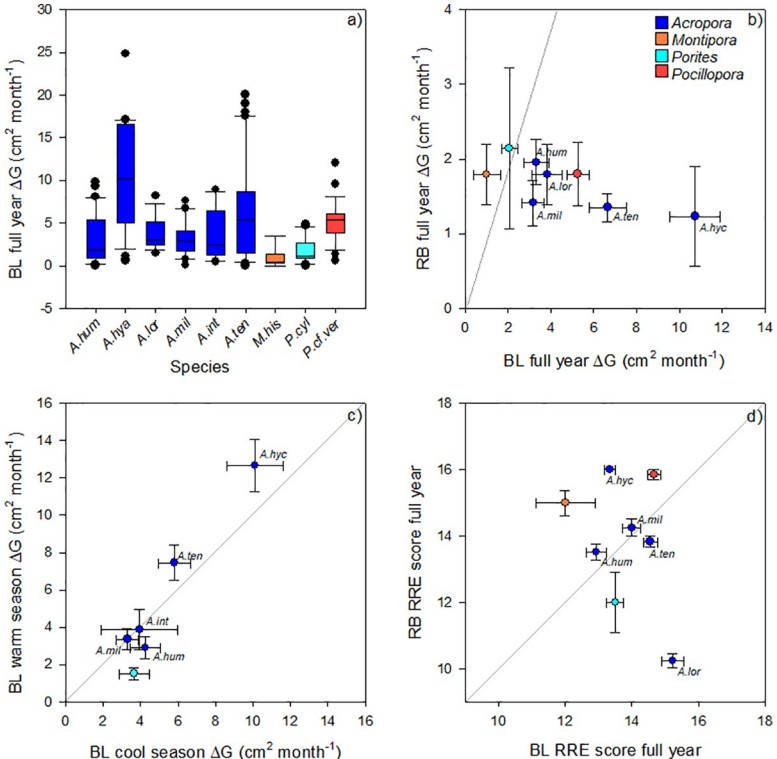

**Fig 2.** Comparisons of **(A)** Growth rates (areal extension; ΔG, cm² month⁻¹) measured over the full year (August 2018-July 2019) for the 9 coral taxa maintained at Blue Lagoon (BL) nursery: *Acropora humilis* (A.hum), *Acropora hyacinthus* (A.hya), *Acropora loripes* (A.lor), *Acropora millepora* (A.mill), *Acropora intermedia* (A.int), *Acropora tenuis* (A.ten), *Montipora hispida* (M.his), *Porites cylindrica* (P.cyl) *Pocillopora* cf. *verrucosa* (P.cf.ver). Box plots for ΔG show the interquartile range (representing 50% of data) and mean (horizontal line)–whiskers extend to the maximum and minimum values of data and excluding outliers (lie outside the 10th and 90th percentiles) **(B)** Mean (± standard error) ΔG of comparative species grown over a full year at BL versus at RayBan (RB; May 2018-February 2019) nurseries; **(C)** Mean (± standard error) ΔG of species grown at BL for data collection periods August-December 2018, "cool season" (Fig 1), versus December 2018-July 2019, "warm season" (as per S4 Table); **(D)** Mean (± standard error) for the RRE score of comparative species grown over a full year at BL versus RB (Table 1). Diagonal line in plots (b-d) signify 1:1.

## Results

At Opal Reef we observed highly variable growth rates (ΔG; cm² month⁻¹) across coral taxa, determined from an entire year of growth at BL (August 2018-July 2019), with highest and lowest values observed for *A. hyacinthus* (10.8 ± 1.2, mean ± standard error) and *M. hispida* (1.0 ± 0.6) (Fig 2A; Table 2) (ANOVA p<0.001; S2 Table). The high ΔG observed for *A. hyacinthus* was almost double that for the next fastest growing coral, *A. tenuis*, and is demonstrated at BL in Fig 3A and 3B. Most other *Acropora* spp. (and *P*. cf. *verrucosa*) exhibited ΔG of ca. 3.6–6.6 cm² month⁻¹, except *A. humilis* and *A. millepora* where ΔG values were lower (3.2–3.3 cm² month⁻¹) and statistically indistinguishable from ΔG for *M. hispida* and *P. cylindrica* (1.0–2.1 cm² month⁻¹, S2 Table).

Likewise, ΔG at Opal Reef site BL were not matched by those for the same species grown within the RB nurseries (Fig 2B; Table 2; S3 Fig). All species at RB exhibited ΔG (mean ± standard error) ranging between 1.2 ± 0.7 cm² month⁻¹ (*A. hyacinthus*) and 2.1 ± 1.1 (*P. cylindrica*), and therefore values markedly lower than those from BL for most species; for example, 1.2 ± 0.7 versus 10.8 ± 1.2 (RB versus BL, *A. hyacinthus*), 1.4 ± 0.2 versus 6.6 ± 0.9 (RB versus BL, *A. tenuis*) and 1.8 ± 0.4 versus 5.3 ± 0.5 (RB versus BL, *P*. cf. *verrucosa*) (Fig 2B, Table 2). However,

**Table 2. Summary data collected from Opal Reef nurseries at sites Blue Lagoon (BL) and RayBan (RB) tracking growth and survivorship over August 2018-July 2019 (BL) and May 2018-February 2019 (RB).**

| Species | Site | Growth | | | Survivorship | | RRE score |
|---|---|---|---|---|---|---|---|
| | | No. | (cm$^2$ month$^{-1}$) | % | No. | % | |
| *Acropora hyacinthus* | BL | 29 | 10.750 (1.174) | 60.40 (9.30) | 216 | 87.6 | 13.34 (0.17) |
| | RB | 3 | 1.232 (0.665) | 58.88 (1.50) | 103 | 97.0 | 16.00 (0.00) |
| *Acropora humilis* | BL | 27 | 3.319 (0.586) | 19.57 (2.92) | 159 | 91.9 | 12.92 (0.30) |
| | RB | 25 | 1.955 (0.302) | 10.90 (1.41) | 116 | 96.3 | 13.52 (0.23) |
| *Acropora intermedia* | BL | 7 | 3.571 (1.272) | 20.19 (8.83) | 19 | 94.7 | 13.14 (0.40) |
| *Acropora loripes* | BL | 9 | 3.837 (0.704) | 24.23 (9.02) | 38 | 100 | 15.22 (0.32) |
| | RB | 32 | 1.346 (0.256) | 7.71 (1.38) | 210 | 79.8 | 10.24 (0.21) |
| *Acropora millepora* | BL | 16 | 3.167 (0.529) | 16.36 (3.44) | 85 | 96.3 | 14.00 (0.27) |
| | RB | 24 | 1.412 (0.304) | 27.61 (10.26) | 183 | 96.8 | 14.25 (0.26) |
| *Acropora tenuis* | BL | 41 | 6.643 (0.866) | 39.12 (7.36) | 372 | 97.4 | 14.55 (0.21) |
| | RB | 6 | 1.353 (0.190) | 13.23 (2.36) | 493 | 98.5 | 13.83 (0.17) |
| *Montipora hispida* | BL | 5 | 0.999 (0.634) | 5.59 (3.91) | 53 | 90.9 | 12.00 (0.89) |
| | RB | 11 | 1.792 (0.410) | 18.45 (3.97) | 26 | 100.0 | 15.00 (0.38) |
| *Montipora spumosa* | RB | 17 | 0.634 (0.263) | 5.02 (1.64) | 57 | 100.0 | 15.84 (0.41) |
| *Pocillopora* cf. *verrucosa* | BL | 26 | 5.253 (0.516) | 30.10 (3.25) | 81 | 98.8 | 14.66 (0.19) |
| | RB | 7 | 1.797 (0.424) | 28.62 (3.45) | 122 | 100.0 | 15.86 (0.14) |
| *Porites cylindrica* | BL | 16 | 2.062 (0.388) | 10.98 (1.88) | 73 | 95.9 | 13.50 (0.26) |
| | RB | 4 | 2.142 (1.078) | 33.99 (22.24) | 31 | 87.1 | 12.00 (0.91) |
| *Turbinaria reniformis* | RB | 18 | 1.668 (0.315) | 16.27 (4.29) | 80 | 100.0 | 14.94 (0.26) |

Data shown are mean (± standard error) for absolute areal growth (cm$^2$ month$^{-1}$) from all tracked fragments (no.); also, the % survivorship for all nursery fragments (no.); and, the mean (± standard error) relative return on effort (RRE) score for the tracked fragments.

site was not identified as a significant factor in testing comparative growth rates between BL and RB (Two-way ANOVA, p<0.001; S2 Table). Only *M. hispida* exhibited ΔG values that were higher at RB than BL. Examples of coral growth at RB are shown in Fig 3C and 3D.

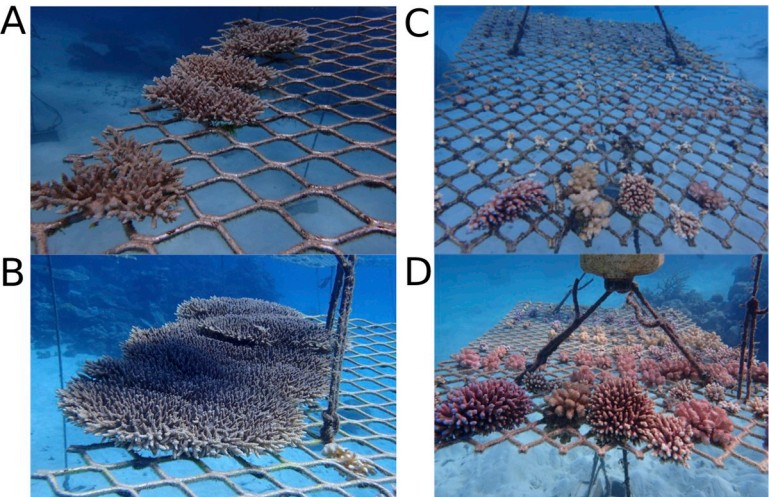

**Fig 3.** Visual examples of coral growth (*Acropora hyacinthus*) on the Blue Lagoon coral nursery from (**A**) 18th October 2018; and then (**B**) 26th August 2019. Examples of coral growth taken from RayBan nurseries from (**C**) 22nd October 2018; versus (**D**) 9th September 2019. Photographs taken by John Edmonsdon.

**Table 3. Summary data collected from Opal Reef nursery sites Blue Lagoon (BL) tracking growth and survivorship across two "seasons", August 2018-December 2018 ("cool") and December 2018-July 2019 ("warm") (see also Fig 1).**

| Species | Season | Growth | | | Survivorship | | RRE score |
|---------|--------|--------|--------|--------|--------|--------|-----------|
| | | No. | (cm$^2$ month$^{-1}$) | % | No. | % | |
| *Acropora hyacinthus* | Cool | 29 | 10.100 (1.508) | 57.06 (10.31) | 216 | 91.7 | 14.14 (0.24) |
| | Warm | | 12.650 (1.406) | 22.96 (2.34) | 198 | 95.5 | 14.34 (0.17) |
| *Acropora humilis* | Cool | 27 | 4.254 (0.818) | 29.74 (7.28) | 159 | 93.7 | 13.19 (0.29) |
| | Warm | | 2.895 (0.603) | 12.85 (2.32) | 149 | 97.8 | 13.33 (0.28) |
| *Acropora intermedia* | Cool | 7 | 3.951 (2.034) | 24.42 (14.17) | 19 | 100.0 | 14.83 (0.65) |
| | Warm | | 3.865 (1.082) | 10.04 (1.63) | 19 | 94.7 | 12.57 (0.20) |
| *Acropora loripes* | Cool | | Not measured in December 2018 | | | | |
| | Warm | | | | | | |
| *Acropora millepora* | Cool | 16 | 3.301 (0.605) | 16.83 (3.54) | 85 | 100 | 15.00 (0.34) |
| | Warm | | 3.354 (0.575) | 14.33 (3.47) | 85 | 96.3 | 13.81 (0.33) |
| *Acropora tenuis* | Cool | 41 | 5.801 (0.881) | 31.34 (4.77) | 372 | 99.5 | 13.95 (0.19) |
| | Warm | | 7.449 (0.940) | 29.68 (4.42) | 370 | 98 | 14.46 (0.21) |
| *Montipora hispida* | Cool | | Not measured in December 2018 | | | | |
| | Warm | | | | | | |
| *Pocillopora* cf. *verrucosa* | Cool | | Not measured in December 2018 | | | | |
| | Warm | | | | | | |
| *Porites cylindrica* | Cool | 16 | 3.653 (0.816) | 18.36 (3.66) | 73 | 97.3 | 14.29 (0.23) |
| | Warm | | 1.519 (0.339) | 5.43 (1.33) | 71 | 98.6 | 12.87 (0.25) |

Data shown are mean (± standard error).

The more intensive sampling frequency at BL enabled us to compare ΔG for the winter-run up to summer ("cool season"), which generally coincided with RB sampling (Fig 1), versus summer-run down to winter ("warm season") (Fig 2C; Table 3). However, of the 6 coral species with comparative seasonal data from BL, ΔG was generally the same for "warm" versus "cool" seasons (no overall seasonal effect, Two-way ANOVA p<0.001; S4 Table). Only *A. tenuis* exhibited a difference for ΔG over time, which was higher for the "warm" (7.5 ± 0.9 cm$^2$ month$^{-1}$, mean ± standard error) than "cool" (5.8 ± 0.9 cm$^2$ month$^{-1}$) season.

Survivorship ranged from 100 to 88% for *A. loripes* to *A. hyacinthus* at BL (August 2018-July 2019) and 100 to 80% for *M. hispida/T. reniformis* to *A. loripes* at RB (May 2018-February 2019) (Table 2). The lower survivorship of *A. loripes* at RB reflected a small localised die-off on one of the nursery platforms.

RRE was subsequently scored (0–20) from parallel measures of (transformed) %growth versus %survivorship [26]. RRE scores for the entire year from the BL nursery were highest for *A. loripes* (15.22 ± 0.32, mean ± standard error), *A. tenuis* (14.55 ± 0.21) and *P.* cf. *verrucosa* (14.66 ± 0.19) and lowest for *A. intermedia* (13.14 ± 0.40), *A. humilis* (12.92 ± 0.30) and *M. hispida* (12.00 ± 0.89) (Fig 2D; Table 2) (ANOVA p<0.001; S2 Table). No differences in season were observed for RRE scores at BL (Table 3; Two-way ANOVA p>0.05, S4 Table). In all cases, these RRE scores reflected taxon-specific differences in ΔG and not survivorship (above; see also S3 Fig).

RRE scores determined for the 9 months on the RB nursery (May 2018-February 2019) exhibited a slightly larger range (10.2–16.0) than for corals at BL (12.0–15.2), noting the highest and lowest values were for different taxa (Fig 2D; Table 2). Specifically, RRE scores at RB were highest for *A. hyacinthus* (16.00 ± 0.01) and *P.* cf *verrucosa* (15.86 ± 0.14), and ca. 25% higher than corresponding RRE scores for these species at BL (Two-way ANOVA p<0.001, S3

Table). In contrast, RRE scores at RB were lowest for species *P. cylindrica* (12.00 ± 0.91) and *A. loripes* (10.24 ± 0.21), and ca. 15–30% lower than corresponding RRE scores for these species at BL. Whilst variation in RRE across species at any one site appears to be driven by differences in growth (above), this between-site variation appears largely driven by differences in survivorship (S3 Fig).

## Discussion

Recent dramatic loss of coral cover on the GBR [11, 13] has led to efforts to evaluate how low-cost nurseries could be used to propagate coral [26] to support high throughput outplanting [27] in this bioregion. Here we report the first 9–12 months of data from multi-taxa nursery installations at a high value reef site on Opal Reef.

The generally higher growth rates for *Acropora* spp. and *Pocillopora* spp. over other species at the Opal Reef nurseries are highly consistent with previous observations from nurseries elsewhere [26]; for example, *Acropora hemprichii*, *Acropora muricata*, *Acropora nasuta* and *P.* cf *verrucosa* versus *P. cylindrica* [24], and *P.* cf *verrucosa* versus non-branching *Montipora* spp. [22], and various *Acropora* spp. and *Pocillopora damicornis* versus *Porites* spp. [16]. Few studies to date have evaluated growth performance of *A. hyacinthus*, or indeed of any plating *Acropora* spp., compared to other coral taxa in nursery settings. Notably, Bongiorni et al. [16] documented $\Delta$G of 1.3 cm$^2$ month$^{-1}$ for *A. hyacinthus*, values that were generally lower than for *A. millepora* (2.0–2.9 cm$^2$ month$^{-1}$) and various other non-plating *Acropora* spp. (1.2–2.8 cm$^2$ month$^{-1}$). Whilst the observations of Bongiorni et al. [16] therefore clearly contrast with the high $\Delta$G we observed for *A. hyacinthus* versus other *Acropora* spp. at BL, another nursery-based study reports higher $\Delta$G for plating *Acropora pharaonis* compared to other non-plating *Acropora* spp. (1.7 versus 0.4–1.5 cm$^2$ month$^{-1}$, [20]). *A. hyacinthus* was the slowest growing of all coral species at RB, and hence a pattern perhaps more consistent with the observations of Bongiorni et al. [16], although this comparison is based on few data available for *A. hyacinthus* at RB (n = 3) compared to BL (n = 29).

A large number of environmental factors influence coral growth, and therefore it is perhaps unsurprising that attempting to benchmark growth performance of corals at the BL nursery with RB, or indeed nursery-based studies elsewhere, is challenging–in particular, given the lack of environmental data typical of nursery studies to date [15], including our current study. Coral species propagated within separate nursery sites of different environmental flow, light, and nutrient regimes exhibit different growth rates, notably where particle fluxes may preferentially enhance the growth of some but not all species [16]. Corals in higher flow sites also appear to exhibit faster growth rates when in nurseries [24], but also following direct outplanting [33], where higher flow rates have been reported to drive an increase in active feeding of corals whilst simultaneously reducing predation rates by corallivorous fish [34]. Such prior observations may therefore in part explain the higher growth rates observed at BL than RB for many species here, given the proximity of BL next to the high-flow channel; however, detailed environmental characterisation will be required to verify this notion. Furthermore, given the low sample sizes of some taxa at either BL or RB, as well as the variation in source colonies used within and between sites (Table 1), it is possible that local differences in environment and genotype [23, 35, 36] influence the outcome that some species grow faster or slower at BL compared to RB.

Temperature is a major factor influencing the growth performance of corals. Even within the GBR, growth rates of key coral species, for example *A. muricata*, *P. damicornis* and *Isopora palifera* [37] and *Acropora nasuta*, *Pocillopora* spp. and *Stylophora pistillata* [38], are highly variable across reef sites, with higher linear extension consistently observed for warmer reefs.

As for other environmental factors, it is currently unclear if the two nursery sites at Opal Reef included in our study are characterised by different temperature regimes, as can occur even over short spatial scales [36]. However, BL versus RB values for ΔG (and survivorship) were retrieved from different time frames, thereby encompassing different temperature exposures (Fig 1B). Specifically, ΔG for RB was determined May 2018-February 2019, when SST is generally coolest but warming, whereas ΔG for BL was collected over the entire year. It is therefore plausible that the generally lower growth rates for RB reflect the generally cooler measurement period.

Comparisons of different seasonal conditions on coral nurseries in Malaysia have shown higher growth rates during warmer months for *A. muricata* [39]. Other studies have commonly reported temporal dynamics in growth over time [16, 22–24]. However, our observations at Opal Reef are in fact consistent with recent suggestions that warmer waters on the GBR may now be constraining summer but not winter coral growth, thereby masking seasonal differences [37]. As such, the differences between RB and BL growth rates presumably stem from factors other than their immediate temperature histories.

Survivorship is routinely used to track population success of species within coral nurseries over time [15]. Across efforts to date, survivorship reported within nurseries often appears high (>80%, [10, 20, 22, 24]; and >70%, [15]), with documented losses of coral often occurring through detachment, as opposed to disease or predation [16, 20]. Our observations are therefore highly consistent with survivorship reported from previous efforts, e.g. *A. millepora* (75–100% over 6 months, [16]), *P.* cf. *verrucosa* and *P. cylindrica* (78–83% and 71–84% over 9 months, [24]), with the exception of high survivorship for *A. hyacinthus* (97% and 88% for RB and BL in our study compared with 56–72% over 6 months, [16]). High survivorship supports the continued use of coral nurseries at Opal Reef, but it is important to view this alongside growth rates for specific coral taxa.

Calculating the RRE was introduced as a means to not only evaluate success across efforts, but also to optimise nursery-based propogation practice [26]. For example, at our Opal Reef nurseries, the higher RRE—due to relatively high survivorship and ΔG—for *A. loripes* (BL) and *P.* cf. *verrucosa* (RB) would suggest that propagation of these species may be better concentrated at just one site. Whilst the higher RRE for *A. hyacinthus*–as a result of higher survivorship but substantially reduced ΔG–at RB might also at face value suggest focusing propagation efforts at this one site, it is important to note that the higher *A. hyacinthus* growth at BL would inevitably yield larger, sexually mature colonies faster [40]. Whilst slower growing but more stress resistant genotypes may carry the greatest value for nursery propagation [36], slow growth potentially carries elevated risk of set-backs via prolonged periods in the nursery. It should also be re-emphasised here that our *A. hyacinthus* data was from only n = 3 (and across a different time frame) than for more comprehensive data from BL, and requires further verification. RRE scores can differ for outplanted versus nursery grown corals [26] and therefore should not be taken as an indication of success for nursery grown corals outplanted to the reef.

Importantly, RRE reflects capacity to gain coral biomass as a result of investment of resources into growth versus survivorship, and hence life-history strategy. Thus, using RRE beyond currently benchmarking 'success scores', for example in governing early decision making to maximise propagation yields, without understanding the nature of the scores should be interpreted with caution. It is important that RRE scores be evaluated relative to the goal of propagation and ideally ultimately modified to include key traits underpinning coral resilience, such as size to reproductive age, fecundity and stress resistance [26]. Whilst our intial data for Opal Reef suggests improved 'success' for some species at one site over another, sustaining taxa at sites with alternate environmental conditions (and RRE scores) may in fact be a useful means to "hedge the bet" of nursery success over time.

## Conclusions

We have shown that the first multi-taxa nurseries deployed on the GBR can return high yields of coral growth and survivorship, and hence high RRE scores. Whilst it is clear that tackling climate change as the underlying cause of degradation to the GBR is a priority [2, 11], our observations here suggest that nurseries benefit local "site stewardship" that collectively is central for improved regional-scale management strategies. Tourism pressures on the GBR are generally considered low (but highly focussed) [12, 26] in relation to other stressors, hence more widespread adoption of "site stewardship" practices by tour operators could boost local coral abundance and diversity at high value tourism sites affected by local impacts, such as tourism pressures, and mass bleaching events. The initial nursery phase of coral propagation appears a useful means to supplement coral material naturally available for site stewardship of high value GBR tourism sites via outplanting programs, assuming nursery maintenance costs remain low (e.g. from natural herbivory here rather than laborious manual cleaning; [31]) and RRE remains high for corals subsequently outplanted [41]. Furthermore, assessing how well such nursery infrastructure can be adopted by other GBR site stewards will be critical to fully resolve the utility and effectiveness as a low-cost site management aid. Growth rate data generated through these nurseries also provides important insight of cross-taxa growth performance that is currently lacking for GBR corals [2, 36], and on-going assessment of coral growth and survival at Opal Reef, as well as other sites on the GBR, will be important to assess the ultimate utility of coral nurseries in boosting coral biomass over space and time.

## Supporting information

**S1 Fig. Examples of nursery platform design deployed at Opal Reef sites RayBan (RB) and Blue Lagoon (BL)–see main text.**
(DOCX)

**S2 Fig. Percentage increase in coral growth (% month-1) areal extension (see methods) for 8 coral species at the two nursery sites (RayBan, Blue Lagoon) at Opal Reef.**
(DOCX)

**S3 Fig. Comparative plots of (ln transformed) %growth and (asin) transformed %survivorship–these 2D plots are used to then score RRE as per Suggett et al. [26].**
(DOCX)

**S1 Table. Hard and total cover cover (% ± standard error) for four sites on Opal Reef (SNO, RayBan, Blue Lagoon, Beautiful Mooring).**
(DOCX)

**S2 Table. ANOVA and *post hoc* Tukey Tests of (i) % Growth month-1, and (ii) RRE, at BL (August 2018-July 2019) binned by species.**
(DOCX)

**S3 Table. Two-way ANOVA and *post hoc* Tukey Tests (p<0.05) of (i) % Growth month-1 and (ii) RRE, binned by site (BL versus RB) and by species (see S1 Table).**
(DOCX)

**S4 Table. Two-way ANOVA and *post hoc* Tukey Tests of (i) % Growth month-1, and (ii) RRE, at BL (August 2018-July 2019) binned by species (see S2 Table) and by "season" ("warm" versus "cool").**
(DOCX)

**S1 Data.**
(XLSX)

## Acknowledgments

The authors wish to express immense thanks to the Great Barrier Reef Marine Park Authority, whose support established the permit for the coral nurseries at Opal Reef (G18/40023.1), as well as staff from Wavelength Reef Cruises (notably Robyn Xuereb and Annabelle Doheny), who have continuously supported the operations and data collection.

## Author Contributions

**Conceptualization:** Emma F. Camp, John Edmondson, David J. Suggett.

**Data curation:** Lorna Howlett, Nicola Henderson, David J. Suggett.

**Formal analysis:** Lorna Howlett, Emma F. Camp, Nicola Henderson, David J. Suggett.

**Funding acquisition:** Emma F. Camp, John Edmondson, David J. Suggett.

**Investigation:** Lorna Howlett, Emma F. Camp, John Edmondson, David J. Suggett.

**Methodology:** Emma F. Camp, John Edmondson, David J. Suggett.

**Project administration:** Emma F. Camp, David J. Suggett.

**Resources:** John Edmondson.

**Supervision:** Emma F. Camp, John Edmondson, David J. Suggett.

**Visualization:** Lorna Howlett, Emma F. Camp.

**Writing – original draft:** Lorna Howlett, Emma F. Camp, David J. Suggett.

**Writing – review & editing:** Emma F. Camp, John Edmondson, Nicola Henderson, David J. Suggett.

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
