## [Decision Letter · Decision Letter 0]

5 Jun 2020

PONE-D-20-11054

Coral growth, survivorship and return-on-effort within nurseries at high-value sites on the Great Barrier Reef

PLOS ONE

Dear Dr. Camp,

Thank you for submitting your manuscript to PLOS ONE. After careful consideration, we feel that it has merit but does not fully meet PLOS ONE’s publication criteria as it currently stands. Therefore, we invite you to submit a revised version of the manuscript that addresses the points raised during the review process.

As you will see from the reviews, the context, limitations and rationale for your study and metrics used needs to be better articulated. While the first reviewer saw general value in your reporting on the first coral nursery attempts on the GBR, reviewer #2 suggests the paper be streamlined, reducing redundancy in text and figures, and importantly, providing better justification for the use and usefulness of the RRE index. Please make sure that all relevant information on that index is given in the manuscript, and don't require the reader to look up another reference. Reviewer #2 also would like to see your approach and results be better linked to the wider literature on coral transplantation, and suggests that restructuring the manuscript to separate results and discussion will serve to better place the study in context. Also, the need for restoration at the particular site should be established better. Note that while Reviewer 2 did not see your results as particularly novel, novelty is not a criterion for acceptance in PLoS ONE.

Both reviewers provide a number of specific recommendations for improving the text, adding some details to the background and rationale of the study as well as to the methods. Their comments, as well as some additional editorial remarks, are given below.

Lastly, please ensure that all underlying data is made available, as pointed out by reviewer #2, as this is a prerequisite for your paper being acceptable for publication.

We look forward to receiving your revised manuscript.

Kind regards,

Sebastian C. A. Ferse, Ph.D.

Academic Editor

PLOS ONE

Additional Editor Comments:

Lines 26-30: Some of the first restoration interventions were aimed at addressing damage from ship-groundings in the Western Atlantic (Florida - see work by Harold Hudson, Bill Precht and others). These predated efforts in response to disease.

Line 35: There are also approaches that provide stable substrate to allow for natural recovery via coral recruitment (artificial reefs) or those that restore environmental conditions so that corals can recover (passive restoration)

Line 143: How were collected fragments kept while being processed on deck? Was there any exposure to air? Shading? A bit more detail on handling would be useful.

Line 270: Please use numbers in square brackets instead of year of publication for in-text references.

Line 309: A. formosa - use the junior synonym, A. muricata, throughout

Line 330: Please use numbers in square brackets instead of year of publication for in-text references.

'Australian & Queensland Governments “Solving the bottleneck of reef rehabilitation through boosting coral abundance: Miniaturising and mechanising coral out-planting” to DJS, EC, JE.

We note that one or more of the authors are employed by a commercial company: Wavelength Reef Cruises

Reviewers' comments:

Reviewer's Responses to Questions

**Comments to the Author**

1. Is the manuscript technically sound, and do the data support the conclusions?

Reviewer #1: Yes

Reviewer #2: Partly

2. Has the statistical analysis been performed appropriately and rigorously? 

Reviewer #1: Yes

Reviewer #2: Yes

3. Have the authors made all data underlying the findings in their manuscript fully available?

Reviewer #1: Yes

Reviewer #2: No

4. Is the manuscript presented in an intelligible fashion and written in standard English?

Reviewer #1: Yes

Reviewer #2: Yes

5. Review Comments to the Author

Reviewer #1: This manuscript describes the first coral reef restoration attempt within the Great Barrier Reef and compares the findings between a site located in the Conservation Park Zone vs a site located in the Marine National Park Zone. Corals were grown within two in situ nurseries which were both initiated during different times in 2018 and also run for different number of months (12 vs 9 months). The authors compare return-on-effort scores which derived from both survivorship and monthly growth rates of the fragments. The authors need to be very clear that this coral reef restoration attempt only covers the first phase of the coral gardening method which is the growing of coral fragments in nurseries. They don’t address the outplanting of those corals to the reef. The manuscript is well written and easy to follow. While it does not target the full process of restoration of Opal Reef, it still provides some interesting results from first experimental attempts to grown corals in nurseries in the GBR. It is somewhat difficult to compare the results from both sites which each other because both nurseries were running for different periods of time and the authors should standardise the period of experiment before they make any comparisons. I’d like to see some recommendations of management of the area or how the approach could be scaled-up. Also, I'd like to know whether there were any considerations with respect to genetic diversity prior to selecting the coral donors.

Specific comments:

LN2- LN22: Abstract: please defer from using abbreviations such as RB, BL, RRE and delta G in the abstract. The abstract does not explain how the return-on-effort scores are being calculated or what they mean. Be clear that the study captures only the coral nursery phase of the coral gardening approach and not the outplanting phase.

LN 11: recalculate the growth rates after 9 months for both locations, so that a direct comparison between both is possible over the same period of time.

LN24: Please write the exact number of coral fragments in the nursery. Using ‘ca.’ is confusing.

LN52-55: this part belongs in the discussion section.

LN91: If the nursery has been located in the Conservation Park Zone, the authors should provide permit + permit number that they must have received from GBRMPA or clarify that the restoration site was not within the permit area. I saw – this comes later in LN143.

I hope your find this assessment fair and useful.

Reviewer #2: Review Questions

1 Partly, because while the growth and survivorship results are straight forward, there is little justification for using RRE. The study is short-term but relevant to evaluating nursery performance. After all, the corals are typically fast-growing and there is no evidence presented that stressors impacted the corals during the course of the study. Little evidence was presented that restoration was needed at the site.

2 The statistics used were overkill for what was a straight forward study. Sometimes less is more. There is lots of redundancy in the Tables and Figures, along with repetition in the narrative. I was unconvinced that RRE is a useful metric. I did not find that the supplemental figures added to the story.

3 Supplemental Materials include summary data, but I couldn't find the raw data.

4 Yes, but copy editing is required. I found the combination of Results and Discussion awkward.

Comments to the authors

Introduction

L28 and throughout the use of e.g. and other qualifiers attached to references are not needed. In general, I didn’t try to copy edit.

L31 Need is now greater based on recent loss of coral because of global warming in 2016/17. Heat waves is a euphemism that could be interpreted as natural events.

L36 Outplanting is conducted in Florida and the Caribbean because recruitment does not occur, or rarely occurs. The authors should provide examples of where recruitment is so low that it impacts “high-value sites.”

L38 semi-continuously = continually

L42 re- (or out-) planting = outplanting or transplanting (pick one then be consistent throughout paper).

L52 The premise that the variety of approaches result in variable results is not supported. Alternatively, the variety of results is a result of what the authors list as different environmental conditions, alternatre coral species and genotypes. The point is that nursery methods generally work well, no matter the technique used to grow the corals. Rationale for the work is overstated by saying uncertainty constrains the capacity to effectively grow corals in nurseries. Methods to grow branching corals are well known. Methods to growth boulder or mounding corals are advancing.

L66 There is debate about using naturally occurring fragments for outplanting because with sites that are depauperate of corals you can end up in a zero-sum situation. Obviously, if there is plenty of source material then collection for fragments isn’t a problem. But if there’s plenty of source material, why restore?

L75 Growth and survivorship are essential metrics for measuring the performance of corals grown in nurseries. The authors additionally calculate a novel scoring method described as “return-on-effort,” but the justification for why RRE is needed in addition to growth and survivorship is lacking. Condition (percent living tissue) is an important metric can affect growth and survivorship, but is not discussed (or included in the study). The authors cite high growth, but they need to place this introductory statement in context to corals from the wild. As stated, the reader is left to assume what high growth means. All corals in both nurseries?

Materials and Methods

L95 The Introduction cites justification for the nursery and I assume eventual outplanting because of coral mortality caused by 2016/17 coral bleaching, but the authors state here that the “exact impact from the 2016/17 coral bleaching event is unknown.” Instead, they site a Pers Obs that mortality was patchy. Also, later (L149) they state that species were chosen for their relatively high abundance. That would seem to diminish the need for restoration. What is the difference between targeted and localized restoration? What is a “key site,” versus a regular site for the installation of a nursery?

L124 Fig 1 Discussion of SST appears in the narrative of this section and the figure caption. Drop in figure caption.

L138 Corals were held at 4-6 m depth on the frames, stated as same depth as from where corals were sourced. That makes sense for RB but not BL that is stated as 8 m depth. Corals could be 4 m shallower on the frames at BL if collected from 8 m, the stated depth of the site.

L156 Authors state that a random subset of corals were monitored for fate-tracking. This is confusing. It appears that a subset were monitored for growth but all corals were monitored for survivorship. Maybe? And with different numbers of fragments to start per species why does it make sense to randomly select among all fragments for fate tracking? At least try to get a balanced design where samples sizes are the same, right? If possible.

LL173 Seeding was conducted in May and August 2018, yet earlier it was stated that RB was installed in February (line 118).

L191 Why would you want to evaluate success as the outcome of growth versus survivorship using RRE? No rationale is provided. Indeed. RRE obscures important information about performance in the nursery because ultimately you want larger numbers and larger fragments that are in good condition. At some point, size can become problematic because fragmentation can occur or the corals can start growing into each other or the weight can sink the frames. This seems to be a metric is search of a problem. The condition (percent live tissue) also matters, but it was not measured.

Results and Discussion

L211 I generally prefer that Results and Discussion are presented separately. That way, the significance and context of the work relative to other studies are presented in one location – the Discussion. As presented, it’s difficult to find the story.

Growth rates within GBR nurseries (Two nurseries at one reef is not the GBR)

L213 Be consistent throughout the paper in the order that information about the two nurseries is presented. In Materials and Methods RB is presented first, then BL. So present RB results first, then BL, then compare them.

Generally, there is an excessive amount of narrative that is also found in Tables 2, 3 and Fig 2. It would be useful to include growth data in Table 2 from wild populations. That way, comparisons with the fragments are all in one place and not scattered in the narrative. RRE score could be a separate table, but I find the metric useless and the paper would not suffer if it was dropped.

Survivorship and Relative Return-on-Effort (RRE)

If survivorship curves are routinely used to track nursery coals, then citations would be useful. They are absent.

RRE values are presented in detail, but their relevance is not clear. Why is it useful to calcualate this metric? What do you get from RRE that is absent from the growth and survivorship data? How can RRE be used to “optimize any given practice?”

The discussion and justification of RRE as a useful metric that appears in lines 365-385 is obscure. For example, the higher RRE (due to high survivorship and growth) “suggests that propagation of these species may be better concentrated at just one site, but only where any transport stress to out-plant sites away for the nursery is negligible.” What evidence do they have that transport stress is relevant to their nurseries at the study reef? What does RRE have to do with this statement? Isn’t this just another way to say, one site is better than the other? Indeed, the authors state that RRE shouldn’t be used as anything other than a “success score.” Indeed, they state that even though one nursery performed better than the other, there are good reasons to hedge bets related to nursery locations. Their reasoning to support RRE as a useful metric is unconvincing.

Conclusions

Site stewardship is identified as a need that is benefited by coral nurseries. The authors state that local impacts (but they don’t identify what they mean by local impacts) could be minimized somehow by coral nurseries while at the same time boosting local coral abundance. Nothing in their study addressed outplanting. Nothing in their study addressed local impacts. The work does present growth data, which is useful, but to date has not been a constraint on outplanting, that I’m aware of anywhere. Coral nurseries are generally productive, with growth rates equal to or greater than what is seen in the wild. The condition and growth of corals in nurseries if often better than seen in the wild, because the corals are up off the bottom in a more vigorous hydrodynamic environment.

6. PLOS authors have the option to publish the peer review history of their article (what does this mean?). If published, this will include your full peer review and any attached files.

Reviewer #1: Yes: Elisa Bayraktarov

Reviewer #2: No

---

## [Author Response · Author response to Decision Letter 0]

6 Sep 2020

We thank you and the Reviewers for the highly constructive feedback of our manuscript entitled Coral growth, survivorship and return-on-effort within nurseries at high-value sites on the Great Barrier Reef; the comments have helped strengthen our manuscript. We have considered all of the suggested edits and have made appropriate changes to the manuscript. These changes are documented in the table below as well as in the track-changed version of the updated manuscript. We have also thoroughly re-proofed the manuscript for any editorial errors. We hope the revised manuscript can now be accepted in PLOS One. All documented edits are track changed in the uploaded word document and have been outlined in the uploaded response letter alongside each Editor and Reviewer comment.

---

## [Decision Letter · Decision Letter 1]

2 Dec 2020

PONE-D-20-11054R1

Coral growth, survivorship and return-on-effort within nurseries at high-value sites on the Great Barrier Reef

PLOS ONE

Dear Dr. Camp,

Thank you for submitting your manuscript to PLOS ONE. After careful consideration, we feel that it has merit but does not fully meet PLOS ONE’s publication criteria as it currently stands. Therefore, we invite you to submit a revised version of the manuscript that addresses the points raised during the review process.

The previous reviewers unfortunately were not available to re-evaluate your manuscript, and the subsequent search for new reviewers has caused considerable delay in the review process, for which I apologize. Your revised manuscript and responses to the previous reviewers have now been assessed by a third reviewer and myself. I find that you have well addressed the previous comments and concerns in your revision, and together with the reviewer only identify a few minor, mostly editorial points, as well as a correction in the results, that remain to be addressed. The reviewer's comments, along with my own, are given below.

We look forward to receiving your revised manuscript.

Kind regards,

Sebastian C. A. Ferse, Ph.D.

Academic Editor

PLOS ONE

Additional Editor Comments (if provided):

Line 19: "complementing", not "complimenting"

Line 25: use plural - "translate"

Line 82: "complements", not "compliments"

Line 82: "assessments of growth or survivorship"

Line 163: "resulted in more fragments on the"

Line 166: suggest to change to "assessed for tracking growth" - the term "fate-tracking" likely led to the irritation by the previous reviewer #2 with this section

Line 191: likewise suggest to simply use "tracked fragments"

Line 203: slightly confusing sentence, suggest rephrasing to "but instead in terms of a potential life-history strategy trade-off"

Line 252 and 254: "tracked fragments"

Line 296: Why is a significant test result reported if no differences were observed? Test result in Table S4 is "ns".

Reviewers' comments:

Reviewer's Responses to Questions

**Comments to the Author**

1. If the authors have adequately addressed your comments raised in a previous round of review and you feel that this manuscript is now acceptable for publication, you may indicate that here to bypass the “Comments to the Author” section, enter your conflict of interest statement in the “Confidential to Editor” section, and submit your "Accept" recommendation.

Reviewer #3: (No Response)

2. Is the manuscript technically sound, and do the data support the conclusions?

Reviewer #3: Yes

3. Has the statistical analysis been performed appropriately and rigorously? 

Reviewer #3: Yes

4. Have the authors made all data underlying the findings in their manuscript fully available?

Reviewer #3: Yes

5. Is the manuscript presented in an intelligible fashion and written in standard English?

Reviewer #3: Yes

6. Review Comments to the Author

Reviewer #3: L6 typo error for the word 'propagated'

L128 change 'rarely subject' to 'rarely subjected'

L157 don't italicized 'cf.' and make it consistent throughout the manuscript

L163 abbreviate A. tenuis

L190 "...any fragments with visible signs of tissue loss were removed" does this mean that even if the tissue loss was very minimal (e.g. 10% loss) for a particular fragment, it was still removed? and what did you do to those fragments? can clarify this in text

L209 Is this mainly to aid the Discussion? If so, then I suggest mentioning this in the Discussion instead, when comparing your results to previous studies

L214 I suggest to briefly mention how to interpret RRE scores

L234 I suggest not to abbreviate those species from different genus but begin with the same letter i.e. Porites cylindrica and Pocillopora cf. verrucosa

L296 Insert 'space' in between A. and humilis

L303 should be written as P. cf verrucosa throughout the manuscript, not P. verrucosa

L312 can provide a range or a percentage loss of coral cover? and do you mean the entire GBR?

L321-325 I suggest to split the statements. Separate the statement - "Notably, Biongiorni et al. ......"

L344 Can you discuss briefly as to why corals exhibited higher growth rates in areas next to high flow channel? can mention other studies that examined the effect of hydrodynamics on coral growth?

L390 typo error for the word 'comprehensive'

L398 add the word 'interpreted' between "should be" and "with caution"

References

1. Journal name should be 'Marine Pollution Bulletin' not 'Marine pollution bulletin'

5. Journal name should be 'Bulletin of Marine Science' not 'Bulletin of marine science'

8. Journal name should be 'Restoration Ecology' not Restoration ecoloy'

10, 17 & 35. italicize Acropora cervicornis

21. Journal name should be 'Ecological Engineering' not 'Ecological engineering'

29. Journal name should be 'BMC Bioinformatics' not 'BMC bioinformatics'

34. Italicize Acropora spp.

37 Journal name should be 'Scientific Reports' not 'Scientific reports'

7. PLOS authors have the option to publish the peer review history of their article (what does this mean?). If published, this will include your full peer review and any attached files.

Reviewer #3: No

---

## [Author Response · Author response to Decision Letter 1]

12 Dec 2020

Editors and reviewers comments have been addressed and responded to within the file labelled "Response to Reviewers" uploaded as the Cover Letter.

---

## [Editor Report · Decision Letter 2]

21 Dec 2020

Coral growth, survivorship and return-on-effort within nurseries at high-value sites on the Great Barrier Reef

PONE-D-20-11054R2

Dear Dr. Camp,

We’re pleased to inform you that your manuscript has been judged scientifically suitable for publication and will be formally accepted for publication once it meets all outstanding technical requirements.

Kind regards,

Sebastian C. A. Ferse, Ph.D.

Academic Editor

PLOS ONE
---

## [Editor Report · Acceptance letter]

2 Jan 2021

PONE-D-20-11054R2 

Coral growth, survivorship and return-on-effort within nurseries at high-value sites on the Great Barrier Reef 

Dear Dr. Camp:

I'm pleased to inform you that your manuscript has been deemed suitable for publication in PLOS ONE. Congratulations! Your manuscript is now with our production department. 

Kind regards, 

on behalf of

Dr. Sebastian C. A. Ferse 

Academic Editor

PLOS ONE